# Acute kidney injury among critically ill neonates in a tertiary hospital in Tanzania; Prevalence, risk factors and outcome

**Naomi A. Mwamanenge, Evelyn Assenga, Francis F. Furia**◯*

Department of Paediatrics and Child Health, School of Medicine, Muhimbili University of Health and Allied Sciences, Dar es Salaam, Tanzania

* Fredrick.francis78@gmail.com

## Abstract

### Background

Neonatal acute kidney injury contributes to high mortality in developing countries. The burden of neonatal AKI is not known in Tanzania despite having high neonatal mortality. This study was conducted to determine the burden of AKI among critically ill neonates admitted at Muhimbili National Hospital.

### Methods

This was a cross-sectional study conducted in the neonatal ward at the MNH. Eligible critically ill neonates were recruited consecutively between October 2017 and March 2018. Data was collected using a standardized structured questionnaire. Blood specimen was drawn to measure baseline creatinine at admission, 48th hour, 72nd hour and 14th day. Data was analysed using SPSS version 20.0 Univariate analysis was done using chi-square to determine the association between categorical variables and multivariate logistic regression was performed to determine predictors of AKI.

### Results

A total of 378 critically ill neonates were recruited, 31.5% had AKI and independent predictors of AKI were noted to be neonatal sepsis (aOR 2.237, 95%CI 1.3–3.6, P = 0.001), severe pneumonia (aOR3.0, 95%CI 1.0–9.3, P = 0.047) and use of gentamycin (aOR6.8, 95%CI 1.3–9.3, P = 0.02). Complete resolution of renal dysfunction at the fourteenth day was seen in 83.1% of the neonates while 16.9% had persistence of renal dysfunction. Ultrasound scan were performed among 105 participants with AKI revealed increased echogenicity, mild hydronephrosis and ectopic kidneys in 25 (23.8%), 4 (3.8%) and 2 (1.9%) respectively. In-hospital mortality was significantly higher among neonates with AKI (70.6%) as compared to those without (29.4%) p< 0.001.

**Data Availability Statement:** All relevant data are within the manuscript and its Supporting Information files.

**Funding:** The authors received no specific funding for this work.

**Competing interests:** The authors declare that no competing interests exist.

## Conclusions

AKI was noted in a third of critically ill neonates, with neonatal sepsis, severe pneumonia and use of gentamycin as independent predictors of AKI. Neonates who suffered AKI had twice as much mortality as compared to those without.

## Introduction

The leading causes of neonatal mortality globally include sepsis, prematurity and birth asphyxia. [1] These three conditions are reported to be important risk factors for neonatal acute kidney injury (AKI). [2–4] Neonatal AKI has been reported to contribute to mortality and increased hospital stay which significantly increase the cost of care posing challenges for countries with limited resources. [5] Most countries in lower and middle income settings including Tanzania have limited facilities and skilled health care providers for provision of care to patients with kidney diseases particularly those requiring renal replacement therapy.[6] International Society of Nephrology through its 0 by 25 initiative declared management of AKI a human right issue promoting elimination of preventable deaths from AKI by 2025. [7, 8]

In Tanzania, 2.1 million babies were born in 2015, and 39,000 neonates are estimated to have died from birth asphyxia (29.3%), prematurity (24.7%) and sepsis (19.7%) which are the leading causes neonatal mortality. [9–11] Prevalence of AKI among neonates is estimated to be between 11% and 56% as reported by various studies globally. [2–5] This study was conducted to uncover the contribution of AKI in neonatal morbidity and mortality in Tanzania, and was aimed at determining the prevalence, risk factors and immediate outcome of AKI among critically ill neonates admitted at Muhimbili National Hospital neonatal ward.

## Methods

### Study site and design

This was hospital based cross-sectional study conducted at the Muhimbili National Hospital (MNH) neonatal unit. MNH which is in Dar es Salaam city with a population of 8 million people is the national referral hospital and is also the teaching hospital for Muhimbili University of Health and Allied Sciences (MUHAS). MNH receives patients from 29 regional referral hospitals in the country three of which are in Dar es Salaam city (Amana, Temeke and Mwananyamala). The neonatal ward at MNH has a capacity of 140 beds, admitting approximately 20–30 babies per day with an annual admission rate of 6000–8000 neonates.

### Inclusion and exclusion criteria

All critically ill neonates admitted in neonatal ward were eligible for this study. Details of the study were provided to parents of all critically neonates admitted; only neonates of parents who consented were recruited. Neonates with obvious congenital abnormalities of genital urinary tract like bladder exstrophy were excluded.

Sample size was estimated using single population proportion formula taking prevalence of AKI (33.3%) reported among admitted neonates in Zimbabwe by Matyanga *et al*., 95% confidence interval, 5% level of significance and maximum allowable error of 0.005. [12]

## Study procedures

Critically ill neonates admitted in the neonatal unit were recruited consecutively until the minimum sample size was reached. A structured questionnaire was used to collect data. Demographic information including gestational age, birth weight, Apgar score, was obtained from interviewing the mother and by reviewing participants' case notes.

The clinical presentation of the neonate was obtained through physical examination and review of patient case notes. The treatment received by the neonate was also assessed to determine if considerations were made regarding renal safety and this was documented. Neonates were followed for 72 hours to determine the progression of their AKI and at 14 days to document their outcome including discharge from the hospital, prolonged hospitalization or death.

Blood specimen was drawn from recruited participants at admission, after 72hours and at the 14th day. The specimen was drawn from the antecubital fossa using aseptic technique; 1 ml of venous blood drawn was stored in red-capped vacutainer and transported to the MNH Central Pathology Laboratory. The specimen was analyzed for serum creatinine using Architect plus Ci4100 analyzer. Other tests that were analyzed were blood urea nitrogen (BUN) full blood picture (FBP), C- reactive protein (CRP) and serum electrolytes. Kidney, ureter and bladder ultrasound scans were performed for participants who developed AKI.

## Study definitions

KDIGO criteria of a rise of serum creatinine of 26.5 μmol/L from baseline within 48 hours or an increase in serum creatinine to 1.5 times baseline, which is known or presumed to have occurred within the prior seven days were considered in this study, and the diagnosis of AKI was established by measuring serum creatinine at baseline and 48 hours after admission. [13]

## Study follow up

All neonates with AKI were followed up whereby their serum creatinine was measured at 48th hour, 72nd hour, 14th day and 28th day to determine resolution of renal dysfunction.

## Study outcome

The primary outcome was occurrence of AKI and the other outcomes included death and duration of hospital stay.

## Data analysis

All questionnaires were checked for consistency. Data entry, cleaning, and analysis were done using Statistical Package for Social Science (SPSS) version 20.0. Continuous variables were expressed using measures of central tendency while categorical data such as serum creatinine values were expressed as frequencies or proportions. Univariate analysis was done using chi-square to determine the association between the different associated factors and AKI. Multivariate logistic regression was then used to determine the association between AKI and known associated factors found to be significant on univariate analysis. A p value of <0.05 was considered statistically significant.

## Ethical consideration

This study was approved by MUHAS Institutional Review Board and permission to conduct this study at MNH was sought from the administration. Signed informed consent was obtained from parents prior to recruitment of participants. Clinical information obtained in

this study was communicated to participants' attending physicians so that they could be used for clinical care.

## Results

### Social-demographic characteristics of the study participants

A total of 378 neonates were enrolled in this study, of which 81% (306/378) were ≤ 7 days in age and 59.8% were male (226/378). Majority of the participants were full term (78.0%), had birth weight above 2500 grams (74.3%) and a five-minute Apgar score above 7 (82.5%). Median age (Inter-Quartile Range) of participants at admission was 3 (1, 6) days, Table 1.

### Prevalence of AKI among critically ill neonates

Prevalence of AKI in this study was 31.5% (119/378), AKI stage I was noted in 78.2% of participants with AKI while stage II and III contributed to 9.2% and 12.6% respectively, Fig 1.

### 4.3 Trend of serum creatinine

Fig 2 describes a scatter diagram showing participants' serum creatinine at baseline and their follow up. The median serum creatinine was 166.5 μmol/L with maximum of 1338 μmol/L. Fig 3 shows box plot of baseline, 72 hours and 14-day serum creatinine levels, for day 14 serum creatinine was performed for participants with AKI.

### Factors associated with AKI

Participants who presented with neonatal sepsis (p = 0.004), intestinal obstruction (0.021), severe pneumonia (p = 0.001), and meningitis (p = 0.049) were noted to have higher occurrence of AKI as compared to those without these conditions, as described in Table 2. Participants who had received gentamycin were also noted to have higher rates of AKI as compared to those who did not receive (p = 0.014).

**Table 1. Socio-demographic characteristics of study participants (n = 378).**

| Demography | N (%) |
|---|---|
| **Age (Days)** | |
| ≤ 7 | 306 (81.0) |
| 8 to 14 | 43 (11.4) |
| 15 to 28 | 29 (7.7) |
| **Sex** | |
| Male | 226 (59.8) |
| Female | 152 (40.2) |
| **Birth weight (g)** | |
| Normal (2500 and above) | 281 (74.3) |
| Low birth weight (1500–2499) | 70 (18.5) |
| Very low birth weight (1000–1499) | 24 (6.3) |
| Extremely low birth weight (<1000) | 3 (0.8) |
| **Apgar score at 5th minute** | |
| ≤ 6 | 66 (17.5) |
| ≥7 | 312 (82.5) |
| **Gestational age** | |
| ≥37 weeks | 295 (78.0) |
| <37 weeks | 83 (22.0) |

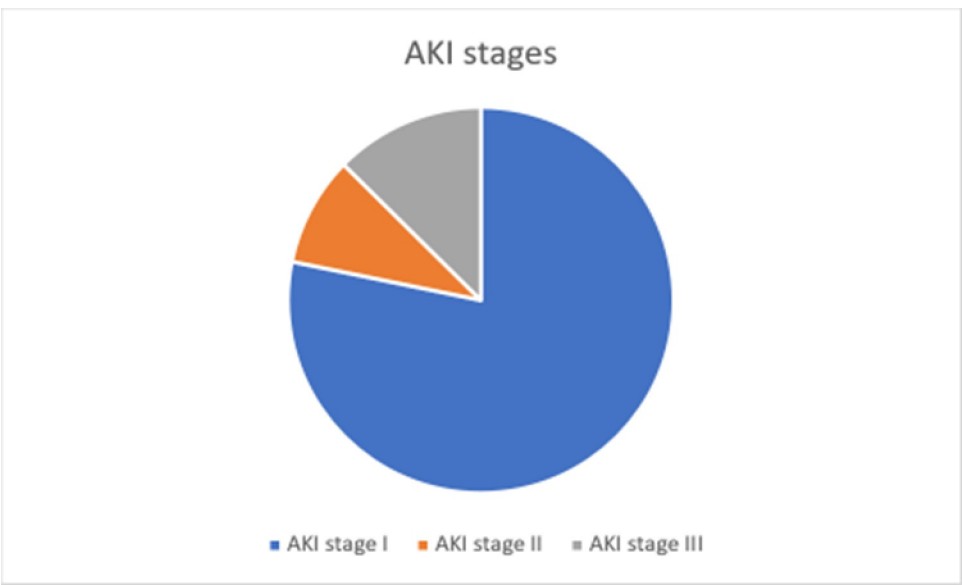

**Fig 1. Pie chart showing AKI stages.**

## Multivariate logistic regression

The following factors were noted to be independent predictors of AKI, neonatal sepsis (aOR 2.237, 95%CI 1.3–3.6, P = 0.001), severe pneumonia (aOR3.0, 95%CI 1.0–9.3, P = 0.047) and usage of gentamycin (aOR6.8, 95%CI 1.3–9.3, P = 0.02) as shown on Table 3.

## Ultrasound findings of participants with AKI

Seventy-four (70.5%) participants with AKI had normal kidney, ureter and bladder ultrasound scans (KUB), twenty-five (23.8%) had increased echogenicity, four (3.8%) had mild hydrone-phrosis and two (1.9%) had ectopic kidneys. Fourteen participants with AKI did not have an ultrasound KUB since they died before 48hours, Fig 4.

## Outcome for patients with AKI

Table 4 describes the outcome of study participants, the overall in-hospital mortality in this study was 22.5% (85/378), and participants with AKI had significantly higher mortality (70.6%) as compared to those without AKI (29.4%) p value of 0.01. Among those with AKI, 62.5% had longer hospital stay of more than fourteen days compared to 37.5% without AKI and p value of 0.006.

Fifty-nine participants who had AKI survived, out of which 83% (49/59) recovered their renal functions within 14 days and 17% recovered within 28 days.

## Discussion

Burden of neonatal morbidity among critically ill neonates in lower- and middle- income countries is usually attributed to neonatal sepsis, prematurity and perinatal asphyxia, these conditions also account for high neonatal mortality. These conditions may lead to short term and long-term complications including acute kidney injury. This study has demonstrated that AKI is very common among critically ill neonates at MNH with an occurrence of 31.5% based on KDIGO criteria. This rate affirms findings from other studies including a multicentre

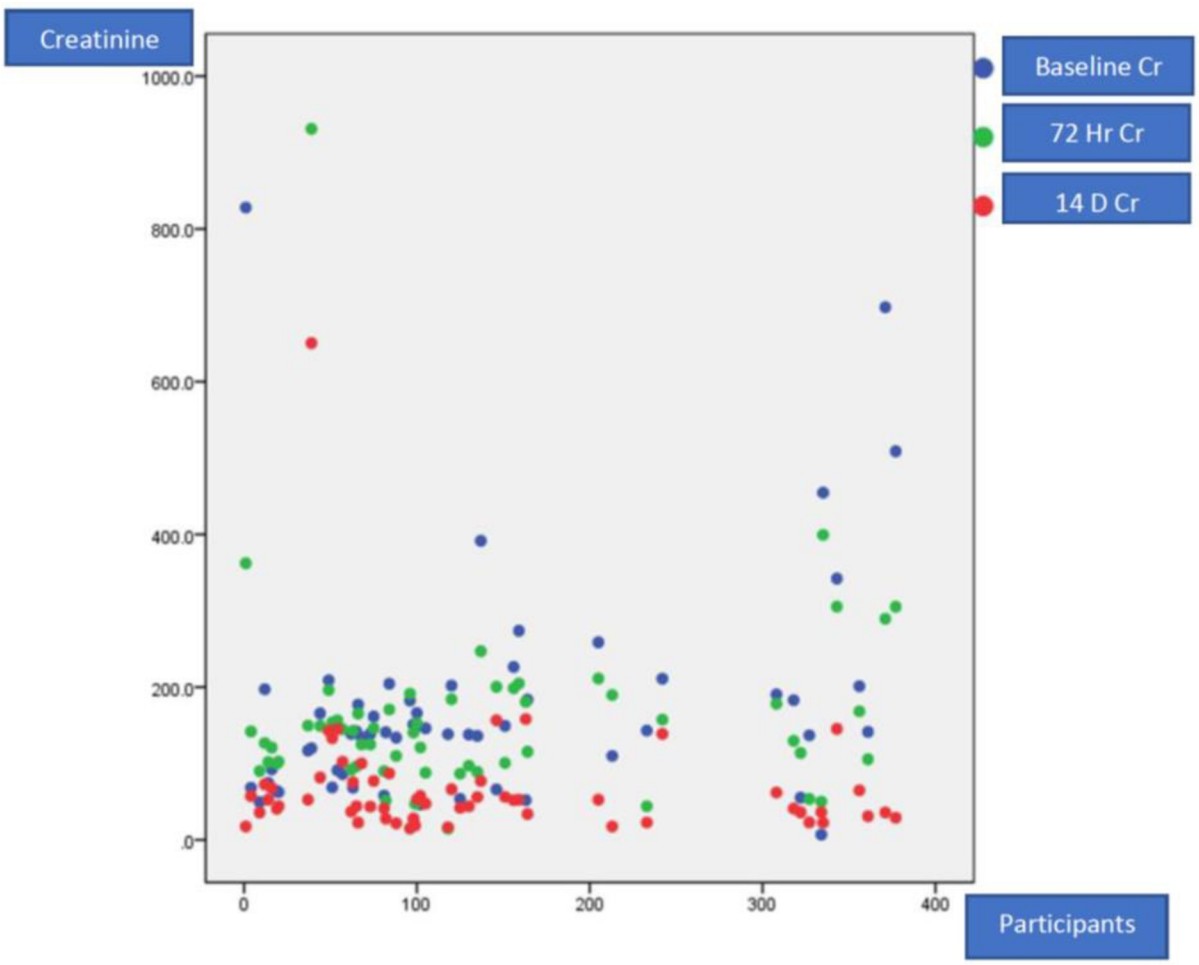

**Fig 2. Trend of serum creatinine at baseline and follow up for study participants.**

(AWAKEN) study which reported a rate of 27% among neonates in intensive care units in developed countries and one conducted by Abdelraheem *et al* in Sudan [5, 14]. Alaro et al reported a significantly lower prevalence (11.7%) of AKI among neonates in a study conducted in Kenya, although the settings are similar, in that study only neonates with birth asphyxia were recruited. [2] In contrast to our findings, Lee et al reported a very high prevalence of AKI among extremely low birth weight neonates admitted in ICU in Taiwan, the higher rate reported by Lee et al compared to our findings could be attributed difference in participants and settings, indicating high vulnerability to AKI faced by extremely low birth weight neonates. [3]

Majority of neonates in this study had AKI stage I (78.2%). The diagnosis of AKI was established by measuring serum creatinine at baseline and 48 hours after admission, despite the caveats of using creatinine, it is important for clinicians to consider measuring serum creatinine at baseline and repeat after 48 hours for all critically ill neonates based on high rate of AKI noted in this study [13,15]. This will facilitate timely diagnosis of AKI and in early stages, which is relatively easy to manage and requiring minimal resources.

Congenital abnormalities of kidney and genital-urinary tract are also reported to contribute to AKI among neonates [16]. In our findings, 29 out of 74 participants with AKI who had

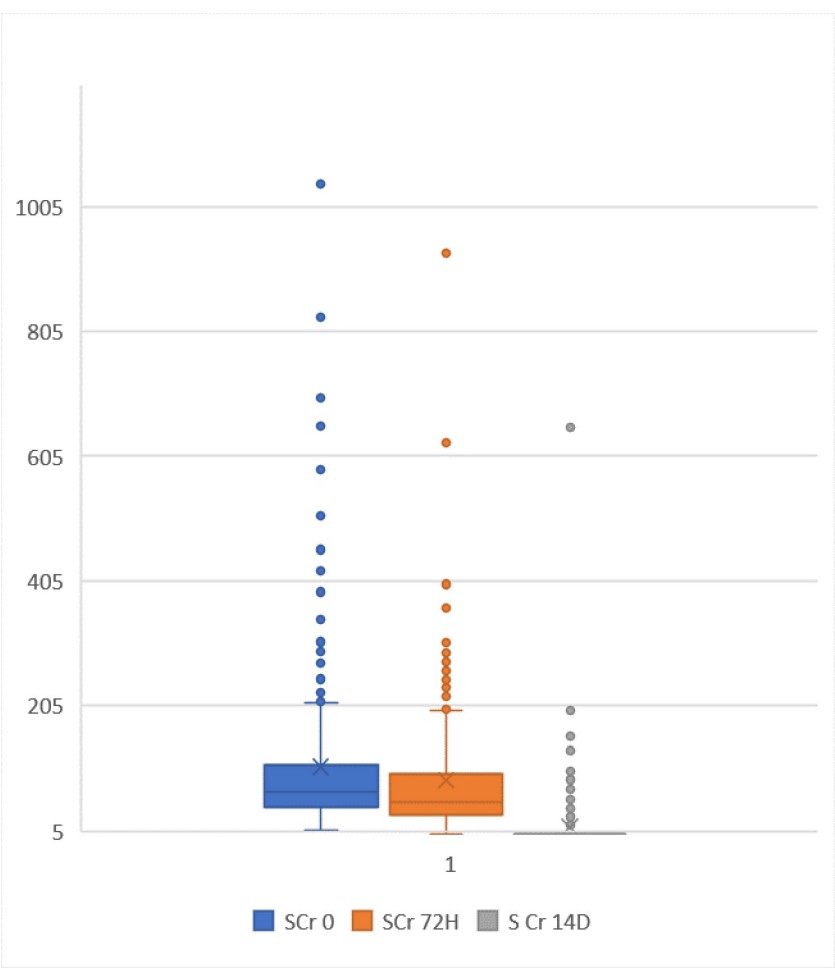

**Fig 3. Box plot showing baseline, 72 hours and 14-day serum creatinine of participants.**

KUB ultrasound scans had congenital abnormalities of the genitourinary system detected only by ultrasound. This highlights the importance of performing ultrasound scanning for neonates with AKI to avoid missing congenital abnormalities most of which if not properly managed, are known to result in end stage renal failure in the first decade of life requiring renal replacement therapy including kidney transplantation.[14, 17, 18]

Congenital abnormalities, hypoxia, infections, nephrotoxic medications, dehydration, and prematurity are known to contribute to AKI among neonates. [4, 5, 19]. In this study neonatal sepsis, meningitis, severe pneumonia, intestinal obstruction, and gentamycin use were noted to be risk factors for occurrence of AKI. Neonatal sepsis, severe pneumonia and gentamycin use were independent predictors of AKI observed in this study. Gentamycin is one of the recommended first line antibiotics for managing neonatal sepsis, therefore it is important for clinicians to be on the alert for AKI when treating critically ill neonates with this drug [20].

Neonatal AKI may result in devastating outcomes including prolonged hospital stays, chronic kidney disease, end stage renal failure and deaths. [5, 14, 16, 21, 22]. Mortality as high as 73% was reported for children with AKI in a systematic review of AKI in sub-Saharan Africa [5]. The overall mortality in this study was 22.5%, neonates with AKI had significantly higher mortality (70.6%) as compared to those without AKI (26.4%). All neonates with AKI in this study were managed conservatively which might have accounted for this high mortality [5].

**Table 2. Factors associated with AKI (n = 378).**

| Variable | With AKI N (%) | Without AKI N(%) | *p*- value |
|---|---|---|---|
| **Age (Days)** | | | |
| $\leq 7$ | 92 (30.1) | 214 (69.9) | |
| 8 to 14 | 18 (41.9) | 25 (58.1) | |
| 15 to 28 | 9 (31.0) | 20 (69.0) | **0.296** |
| **Sex** | | | |
| Male | 69 (30.5) | 157 (69.5) | |
| Female | 50 (32.9) | 102 (67.1) | **0.628** |
| **Gestational age** | | | |
| $\geq$37 weeks | 90 (30.5) | 205 (69.5) | |
| <37 weeks | 29 (34.9) | 54 (65.1) | **0.443** |
| **Neonatal sepsis** | | | |
| Yes | 32 (22.4) | 111 (77.6) | |
| No | 87 (36.9) | 149 (63.1) | **0.004** |
| **Intestinal obstruction** | | | |
| Yes | 18 (60.0) | 12 (40.0) | |
| No | 101 (29.0) | 247 (71.0) | **0.021** |
| **Severe pneumonia** | | | |
| Yes | 4 (16.7) | 20 (83.3) | |
| No | 115(32.5) | 239 (67.5) | **0.001**[*] |
| **Congenital Heart Disease** | | | |
| Yes | 3 (75.0) | 1 (25.0) | |
| No | 116(31.0) | 238 (69.0) | **0.102**[*] |
| **Prematurity with RDS[β]** | | | |
| Yes | 19 (32.8) | 39 (67.2) | |
| No | 100 (31.3) | 220 (68.8) | **0.820** |
| **Meningitis** | | | |
| Yes | 6 (60.0) | 4 (40.0) | |
| No | 113 (30.7) | 255 (69.3) | **0.049**[*] |
| **HIE[α]** | | | |
| Yes | 37 (33.9) | 72 (66.1) | |
| No | 82 (30.5 | 187 (69.5) | **0.512** |
| **Gentamycin used** | | | |
| Yes | 113 (30.5) | 257 (69.5) | |
| No | 6 (4.7) | 121 (95.3) | **0.014** |

Intestinal obstruction included those with hypertrophic pyloric stenosis, duodenal atresia, anal atresia and trachea esophageal fistula.

[*]Fischer exact test was used

[β]Respiratory Distress syndrome

[α]Hypoxic ischaemic encephalopathy

This high mortality depicts the significant contribution of AKI in neonatal mortality in many lower income countries and calls for efforts to build capacity for appropriate management of children with AKI especially provision of renal replacement therapy. [6].

Consistent with other global and regional reports neonates with AKI in this study had prolonged hospital stay as compared to those without. [5] Prolonged hospital stay has serious

**Table 3. Logistic regression on risk factors for AKI (n = 378).**

| Risk factor | Crude OR (95% CI) | p-value | Adjusted OR (95% CI) | p-value |
|---|---|---|---|---|
| **Neonatal sepsis** | | | | |
| Yes | 2.007 (1.249–3.225) | | 2.237 (1.375–3.637) | |
| No | 1 | 0.004 | 1 | 0.001 |
| **Intestinal Obstruction** | | | | |
| Yes | 2.793 (1.171–6.661) | | 0.492 (0.200–1.211) | |
| No | 1 | 0.001 | 1 | 0.123 |
| **Meningitis** | | | | |
| Yes | 3.385 (0.937–12.228) | | 0.4002 (0.106–1.517) | |
| No | 1 | 0.049 | 1 | 0.178 |
| **Severe Pneumonia** | | | | |
| Yes | 3.668 (1.705–7.893) | | 3.083 (1.017–9.399) | |
| No | 1 | 0.106 | 1 | 0.047 |
| **Congenital heart disease** | | | | |
| Yes | 6.672 (0.687–8.428) | | 0.199 (0.020–1.951) | |
| No | 1 | 0.060 | 1 | 0.166 |
| **Gentamycin used** | | | | |
| Yes | 6.823 (1.356–8.423) | | 6.830 (1.321–9.399) | |
| No | 1 | 0.007 | 1 | 0.022 |

implications in the cost of care particularly for developing countries with limited resources. Neonates with AKI who survived in this study recovered their renal functions, recovery was noted within two weeks for majority of these neonates. These neonates are at risk of

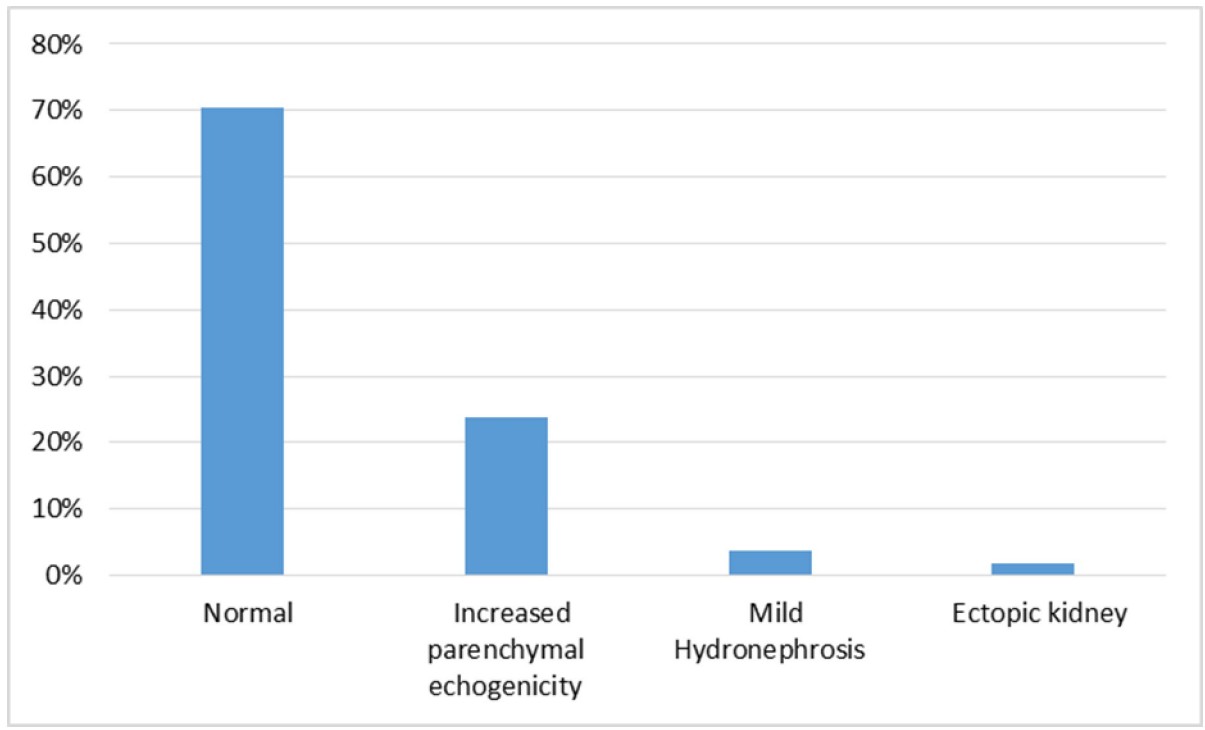

**Fig 4. Distribution of findings on KUB ultrasound of 119 participants with AKI (n = 105).**

**Table 4. Outcome for patients with AKI (n = 119).**

| Outcome | | | p- value |
|---|---|---|---|
| | **With AKI** | **Without AKI** | |
| **Mortality** | | | |
| Dead | 60 (70.6) | 25 (29.4) | |
| Alive | 59 (20.1) | 234 (79.9) | <**0.001** |
| **Hospital stay** | | | |
| ≥14 | 10 (62.5) | 6 (37.5) | |
| <14 | 109 (30.1) | 253 (69.9) | **0.006** |

developing chronic kidney diseases including end stage renal failure in future as a result of renal insult caused by AKI. [14].

This is the first study in Tanzania reporting on the burden of AKI among neonates, and the findings of this study have sensitized clinicians working in the neonatal unit at MNH and management of neonates with AKI has seen significant improvement including establishment of acute peritoneal dialysis. A large limitation of this study is possibility of overestimation of the prevalence of AKI as MNH is a tertiary referral hospital. Other limitations included failure to use urine output as a criterion for diagnosing AKI because of difficulties of monitoring, limited follow up period of up to 28 days which means long term complications could not have been ascertained and single centre-based study with limited generalizability.

In conclusion, we conducted a cross-sectional study among critically ill neonates at Muhimbili National Hospital in Dar es Salaam, Tanzania. Nearly 32% of critically ill neonates had AKI with sepsis, severe pneumonia and gentamycin observed to be independent predictors of AKI. Significantly higher mortality (70.6%) was noted among those who suffered AKI. It is therefore important for clinicians caring for critically ill neonates to have a high index of suspicion for AKI.

## Supporting information

**S1 Dataset. Data set for AKI study.**
(XLS)

## Acknowledgments

Authors are grateful to the nurses who assisted with recruitment of participants and to all parents who agreed to let their children to be enrolled in this study.

## Author Contributions

**Conceptualization:** Naomi A. Mwamanenge, Evelyn Assenga, Francis F. Furia.

**Data curation:** Naomi A. Mwamanenge, Evelyn Assenga, Francis F. Furia.

**Formal analysis:** Naomi A. Mwamanenge.

**Methodology:** Evelyn Assenga, Francis F. Furia.

**Supervision:** Evelyn Assenga, Francis F. Furia.

**Writing – original draft:** Francis F. Furia.

**Writing – review & editing:** Evelyn Assenga.

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
