## [Decision Letter · Decision Letter 0]

2 Dec 2019

PONE-D-19-30433

Acute Kidney injury among neonates in a tertiary hospital in Tanzania; prevalence, risk factors and outcome

PLOS ONE

Dear Dr Francis F Furia,

Thank you for submitting your manuscript to PLOS ONE. After careful consideration, we feel that the manuscript has merit but, at in its present form, does not fully meet PLOS ONE’s publication criteria . Therefore, after consideration of Reviewer 1 points, we invite you to submit a revised version of the manuscript that addresses each of the points raised during the review process.

ACADEMIC EDITOR: :

Could you address each of the points individually and return a marked up copyThe manuscript has merit because of the rarity of the information. if you are able to answer each point, we'd like to see a revised version. i agree with the points raised through the excellent review provided below

We would appreciate receiving your revised manuscript by 14/01/2020. To enhance the reproducibility of your results, we recommend that if applicable you deposit your laboratory protocols in protocols.io, where a protocol can be assigned its own identifier (DOI) such that it can be cited independently in the future. For instructions see: http://journals.plos.org/plosone/s/submission-guidelines#loc-laboratory-protocols

We look forward to receiving your revised manuscript.

Kind regards,

David S Gardner

Academic Editor

PLOS ONE

Journal Requirements:

2. Please amend either the title on the online submission form (via Edit Submission) or the title in the manuscript so that they are identical.

3. Please remove your figures from within your manuscript file, leaving only the individual TIFF/EPS image files, uploaded separately.  These will be automatically included in the reviewers’ PDF.

Additional Editor Comments (if provided):

Reviewers' comments:

Reviewer's Responses to Questions

**Comments to the Author**

1. Is the manuscript technically sound, and do the data support the conclusions?

Reviewer #1: Yes

2. Has the statistical analysis been performed appropriately and rigorously? 

Reviewer #1: Yes

3. Have the authors made all data underlying the findings in their manuscript fully available?

Reviewer #1: Yes

4. Is the manuscript presented in an intelligible fashion and written in standard English?

Reviewer #1: Yes

5. Review Comments to the Author

Reviewer #1: This manuscript provides the prevalence, risk factors and outcomes of acute kidney injury among 378 critically ill neonates in a single center tertiary hospital in Tanzania. The authors should be commended for this extensive research undertaking in setting of a resource limited setting. The data presented would be improving upon the available epidemiological literature on neonatal acute kidney injury in East Africa. The ultrasound findings should be further highlighted in this manuscript given that the literature on this topic is rather sparse in developing countries. While the data presented is unique, there are certain points that need to be clarified especially regarding the research methodology, which are delineated below:

Major points:

-To strengthen the introduction, the second paragraph could include what the average international estimated prevalence of AKI is and what is known about neonatal AKI in developing countries with similar limited facilities. Alternatively, the authors may indicate how neonatal rates of mortality from asphyxia, prematurity and sepsis differ in Tanzania compared to global outcomes to provide better context for the manuscript findings.

- Page 4, Line 76-78: “All critically ill neonates admitted in neonatal ward presenting with convulsion, altered level of consciousness, inability to feed, vomiting everything, respiratory distress, jaundice, hyper/hypothermia, dehydration and shock were eligible for this study.” Is it necessary to specify all diagnosis included if the only those with obvious congenital abnormalities of GU tract were excluded? Were there other diagnoses that were excluded?

- Page 4, Line 82-83: “Sample size was estimated using prevalence of AKI (33.3%) reported among admitted 83 neonates in Zimbabwe by Matyanga et al. [12]” Explain how this reference was used to determine sample size for this cross-sectional study.

- Please specify if there were any missing data especially surrounding creatinine. Please specify whether AKI definition was comparing just baseline + 48 hours after admission creatinine in the methods. In general, KDIGO guidelines recommend serum creatinine rise 1.5X baseline within 7 days but creatinine at 7 days did not seem to be included in the study. There also seems to be missing data regarding participants who got KUB Ultrasounds (74+14(died) = 88, but 119 had AKI).

- Indicate in results that no patients were on dialysis.

- Please specify how neonatal sepsis is defined? Should it be neonatal bacteremia instead? In generally, neonatal sepsis could occur from pneumonia, meningitis, etc. Where those with severe pneumonia and meningitis also considered to have neonatal sepsis?

- Specify whether team focused on only in-hospital death.

- Page 15, Line 210-214: “The diagnosis of AKI was established by measuring serum creatinine at baseline and 48 hours after admission” should be included in the methods.

- The authors may want to highlight the amount of congenital abnormalities observed among those with AKI in abstract detected via ultrasound and how it relates to global rates in the discussion.

-Fig 1: Please indicate the distribution of AKI stages

-Fig 2: Consider cleaning up the legend and consider 3 vertical box + whiskers plots to display the distribution of baseline, 72 hour and 14 day creatinine values for all patients to present the data more succinctly.

- The discussion seems to indicate what is consistent in this study with prior data but it would be better if it could also highlight what is different regarding these findings compared to prior findings and reasons for the differences.

- The authors may consider adding the following: A large limitation of this study is that the Muhimbili National Hospital appears to be a secondary/tertiary referral hospital, which likely would lead to overestimation of the prevalence of neonatal acute kidney injury.

Minor points:

- Page 3, Line 52: 0 by 25 Initiative is to eliminate preventable deaths from AKI worldwide by 2025.

- Minor inconsistencies regarding whether MNH is a secondary or tertiary referral hospital.

- Page 7, Line 137-138: Consider: “A total of 378 neonates were enrolled in this study, of which 81% (306/378) were ≤ 7 days in age and 59.8% were male (226/378).”

- Page 7, Line 139: Consider: “Majority of the participants were full term (78.0%), had birth weight above 2500 grams (74.3%) and a five-minute Apgar score above 7 (82.5%).”

- Table 2: define RDS and HIE

- Page 11, Line 189-191: Table 4 describes the outcome of study participants, the overall mortality in this study was 22.5% (85/378), and participants with AKI had significantly higher mortality (70.6%) as compared to those without AKI (29.4%) p value of 0.000� P- value should be < 0.01?

- Page 12, line 204- Missing period

- Page 13, Line 224 and 225: missing comma

- Page 13, Line 229: clinicians

-Page 14, Line 228: missing period

6. PLOS authors have the option to publish the peer review history of their article (what does this mean?). If published, this will include your full peer review and any attached files.

Reviewer #1: No

---

## [Author Response · Author response to Decision Letter 0]

24 Jan 2020

Response to Reviewer #1 comments 

MAJOR POINTS:

To strengthen the introduction, the second paragraph could include what the average international estimated prevalence of AKI is and what is known about neonatal AKI in developing countries with similar limited facilities. Alternatively, the authors may indicate how neonatal rates of mortality from asphyxia, prematurity and sepsis differ in Tanzania compared to global outcomes to provide better context for the manuscript findings.

-A statement has been added to include the average international estimated prevalence of AKI.

Page 4, Line 76-78: “All critically ill neonates admitted in neonatal ward presenting with convulsion, altered level of consciousness, inability to feed, vomiting everything, respiratory distress, jaundice, hyper/hypothermia, dehydration and shock were eligible for this study.” Is it necessary to specify all diagnosis included if the only those with obvious congenital abnormalities of GU tract were excluded? Were there other diagnoses that were excluded?

-This has been changed as recommended, and the work critically ill neonates has been used and the list of other diagnoses omitted.

Page 4, Line 82-83: “Sample size was estimated using prevalence of AKI (33.3%) reported among admitted 83 neonates in Zimbabwe by Matyanga et al. [12]” Explain how this reference was used to determine sample size for this cross-sectional study.

-Explanation of how sample size was obtained has been provided.

Please specify if there were any missing data especially surrounding creatinine. 

-Serum creatinine on day 14 was performed for participants with AKI only, therefore participants without AKI have no values for creatinine for day 14.

Please specify whether AKI definition was comparing just baseline + 48 hours after admission creatinine in the methods. In general, KDIGO guidelines recommend serum creatinine rise 1.5X baseline within 7 days but creatinine at 7 days did not seem to be included in the study.

- The diagnosis of AKI was established by measuring serum creatinine at baseline and 48 hours after admission. This has been included in the methods section 

There also seems to be missing data regarding participants who got KUB Ultrasounds (74+14(died) = 88, but 119 had AKI).

Ultrasound were performed in 

Indicate in results that no patients were on dialysis.

-This has been indicated in the results section.

Please specify how neonatal sepsis is defined? Should it be neonatal bacteremia instead? In generally, neonatal sepsis could occur from pneumonia, meningitis, etc. Where those with severe pneumonia and meningitis also considered to have neonatal sepsis?

-Sepsis was used for neonates with severe infection with no defined focus of infection to differentiate with pneumonia and meningitis

Specify whether team focused on only in-hospital death.

-This study focused on in-hospital deaths only

Page 15, Line 210-214: “The diagnosis of AKI was established by measuring serum creatinine at baseline and 48 hours after admission” should be included in the methods.

-This has been added in the methods section as recommended.

- The authors may want to highlight the amount of congenital abnormalities observed among those with AKI in abstract detected via ultrasound and how it relates to global rates in the discussion.

-This has been included in the abstract as recommended

Fig 1: Please indicate the distribution of AKI stages

-Indicated as recommended

Fig 2: Consider cleaning up the legend and consider 3 vertical box + whiskers plots to display the distribution of baseline, 72 hour and 14 day creatinine values for all patients to present the data more succinctly.

-Changes made as indicated and box plot included in the results section

The discussion seems to indicate what is consistent in this study with prior data but it would be better if it could also highlight what is different regarding these findings compared to prior findings and reasons for the differences.

-The contrast in findings with other studies has been included in the discussion section as recommended.

The authors may consider adding the following: A large limitation of this study is that the Muhimbili National Hospital appears to be a secondary/tertiary referral hospital, which likely would lead to overestimation of the prevalence of neonatal acute kidney injury.

-Added as recommended

MINOR POINTS:

Page 3, Line 52: 0 by 25 Initiative is to eliminate preventable deaths from AKI worldwide by 2025.

-This has been amended 

Minor inconsistencies regarding whether MNH is a secondary or tertiary referral hospital.

-This has been clarified

Page 7, Line 137-138: Consider: “A total of 378 neonates were enrolled in this study, of which 81% (306/378) were ≤ 7 days in age and 59.8% were male (226/378).”

-Changes made as recommended

Page 7, Line 139: Consider: “Majority of the participants were full term (78.0%), had birth weight above 2500 grams (74.3%) and a five-minute Apgar score above 7 (82.5%).”

-Changes made as recommended

Table 2: define RDS and HIE

-RDS and HIE defined 

Page 11, Line 189-191: Table 4 describes the outcome of study participants, the overall mortality in this study was 22.5% (85/378), and participants with AKI had significantly higher mortality (70.6%) as compared to those without AKI (29.4%) p value of 0.000� P- value should be < 0.01?

-Amended as recommended

Page 12, line 204- Missing period

-Period added

Page 13, Line 224 and 225: missing comma

-Comma added

Page 13, Line 229: clinicians

-Changed 

Page 14, Line 228: missing period

-Period added

---

## [Editor Report · Decision Letter 1]

30 Jan 2020

Acute Kidney injury among critically ill neonates in a tertiary Hospital in Tanzania; prevalence, risk factors and outcome

PONE-D-19-30433R1

Dear Dr. Furia,

We are pleased to inform you that your manuscript has been judged scientifically suitable for publication and will be formally accepted for publication once it complies with all outstanding technical requirements.

With kind regards,

David S Gardner

Academic Editor

PLOS ONE

Additional Editor Comments (optional):

i am happy that all comments have been addressed to a rigorous review and that the information in the manuscript adds new information to the literature on AKI in neonates
---

## [Editor Report · Acceptance letter]

5 Feb 2020

PONE-D-19-30433R1 

Acute Kidney injury among critically ill neonates in a tertiary Hospital in Tanzania; prevalence, risk factors and outcome 

Dear Dr. Furia:

I am pleased to inform you that your manuscript has been deemed suitable for publication in PLOS ONE. Congratulations! Your manuscript is now with our production department. 

With kind regards,

on behalf of

Dr. David S Gardner 

Academic Editor

PLOS ONE